# Analysis of the Impact of Environmental and Agronomic Variables on Agronomic Parameters in Soybean Cultivation Based on Long-Term Data

**DOI:** 10.3390/plants11212922

**Published:** 2022-10-30

**Authors:** Elżbieta Wójcik-Gront, Dariusz Gozdowski, Adriana Derejko, Rafał Pudełko

**Affiliations:** 1Department of Biometry, Institute of Agriculture, Warsaw University of Life Sciences, Nowoursynowska 159, 02-776 Warsaw, Poland; 2Departmentof Bioeconomy and Systems Analysis, Institute of Soil Science and Plant Cultivation—State Research Institute (IUNG-PIB), Czartoryskich 8, 24-100 Pulawy, Poland

**Keywords:** yield variability, multivariate analysis, regression trees

## Abstract

Soybean (*Glycine max* (L.) Merr.) is a species of relatively little economic importance in Central and Eastern Europe, including Poland. Due to its popularity for the production of soybean oil, livestock feed, and human food, soybeans are a widely cultivated agricultural crop in the world. The aim of the presented research is to determine the most important agronomic and environmental variables in soybean production in Central and Eastern Europe. This work used a dataset from the Polish Post-Registration Variety Testing System in multi-environmental trials from the years 2012–2021. Variables classified for crop management included doses of mineral fertilizers (N, P, and K) and herbicides, sowing, and the type of previous crops. The environment was also included in the analysis through soil and weather characteristics using climatic water balance (CWB). The analysis was performed using multiple linear regression models and regression trees. It found that the variability of the soybean yield depended mainly on water available to plants and physical soil properties. This means that environmental variables have a stronger effect in comparison to crop management variables. The effect of the nutrients applied in the fields was relatively weak and only important in the case of phosphorus. Other variables which characterize crop management (including sowing date, previous crop, and plant protection using pesticides) have a weak effect on grain yield and yield-related traits variability. As there are not many studies on soybean cultivation in Poland, this work might be used as an introduction to research on soybean management in a hemiboreal climate.

## 1. Introduction

On a global scale, soybean (*Glycine max* (L.) Merr.) is the most important vegetable protein source for food and feed because of its high protein and oil content [1]. The huge importance of this species in the European Union member states, including a growing number of foreign varieties on the domestic seed market in Poland, resulted in an increased interest in soybean and a growth in the number of varietal tests. In the US, in high-yield environments with irrigation, soybean crops produces an average of 4.5 Mg ha^−1^ [2]. In experiments conducted in Argentina [3] and China [4,5], soybean yields reached about 9 Mg ha^−1^. The potential yield of soybean is estimated up to 11 Mg ha^−1^ [6]. In Poland, such high-yield values are rather unattainable due to the hemiboreal climate, with insufficient amount of rainfall, too low temperatures, and the predominance of soils with low fertility. These plants grow wild in the tropics and in the temperate regions of the northern hemisphere. Soybean is a thermophilic plant; thus, high temperatures and the length of daylight have a positive effect on the growth and development of these plants. In the last couple of years in Poland, prolonged drought has been occurring, but when rainfall is present during the flowering period, it has a positive effect on pod formation and seed filling. In 2022, according to ARMA (The Agency for Restructuring and Modernisation of Agriculture), the soybean cultivation acreage in Poland was 48,194 ha (in 2021—25,547 ha).

The cultivation of this species arouses considerable interest among farmers in Poland. This is due to the high demand for soybean meal which is used in the production of feed for farm animals. There is a shortage of specialized soybean processing companies in Poland and the country is still mainly an importer of soybeans. However, over time Poland should also become its producer. The Polish government is planning to reduce soybean imports to be independent of external markets, such as Argentina, Brazil, and the USA, which have the largest soybean production in the world [7]. Polish authorities are promoting the production of alternative sources of domestic protein that can be used to produce high-quality feed [8]. The support of the European Union, which introduced an area payment for the area of leguminous crops, is also of great importance and contributed to the increased interest in growing soybeans. Until recently, attempts to grow soybean in Poland were rather unsuccessful, and therefore the production of this crop was not popular. However, the observed climate changes may create more favorable conditions for growing soybeans in Poland and increase the cultivation area. Due to the improvement of soybean production technology and increases in demand, soybean might become an important arable crop in Poland. Thus, an attempt has been made to investigate optimal growing conditions for Polish soybean cultivation based on available data.

Among many factors, the key decision affecting crop yield is the right dose of fertilizers. The most important nutrients for crops are nitrogen (N), phosphorus (P), and potassium (K) used in mineral fertilization. This also applies to soybean, a plant that is popular in the world but still little known in Poland. However, nitrogen fertilization is of less importance in soybean cultivation due to the symbiosis between plants and nodule bacteria [9]. The availability of nutrients for soybean depends on soil moisture, and high-yield demands optimal soil moisture, especially during flowering and pod-fill stages [10]. The gap between water-non-limiting and water-limiting potential yields is usually very large with low rainfall [11]. Supplemental irrigation at flowering is beneficial for the yield of soybean in water-limited environments [12]; however, in Poland, it is very rarely applied because of environmental and economic constraints [13]. Crop yields, including soybean, depend on physicochemical soil properties of which one of the most important is soil texture (e.g., clay and sand content) [14] and the content of soil organic matter [15]. Loam and clay loam soil texture classes are highly suitable soil for soybean cultivation [16]. Soybean yield in various environmental conditions can be determined by various crop management factors including plant protection [17,18], sowing date [19,20], and application of inoculum [21]. In the study of Zhang et al. [22] relative contribution of agricultural practice was stronger in comparison to climate change. The results were based on the data obtained in China in the years 1981–2010. In the study of Vann et al. [23] conducted in North Carolina (US), the most important crop management factors for soybean yield were maturity group, application of fungicide, and sowing date. Herbicide and insecticide use, irrigation, inoculant seed treatments, tillage, and sowing density were less important for soybean yield variability. Other studies on soybean conducted in the western US corn belt [24] proved a strong effect of water availability (rainfall or irrigation) and sowing date on grain yield. Soybean yield linearly decreased with sowing date delay and it was a very important crop management factor that determine yield variability. 

The yield is one of the most important agronomic parameters and is determined by various yield-related traits. The mass of a thousand grains is one of the basic components of the yield. Plant height affects the number and placement of pods. Taller plants make it easier to harvest seeds and reduce harvest losses. However, plants that are too tall are sensitive to lodging before harvest. The present study focuses on management and environment-related variables in Central and Eastern European soybean cultivation by examining the ranking of the importance of these variables in determining the seed yield, plant height, and 1000-grain weight.

## 2. Results

The total number of observations used in the analysis was 173 (out of 370 possible combinations). The highest number of observations were registered in 2021 and 2020. They amounted to 22.0% and 20.2% of observations in each of these years, respectively. The average soybean yield, with the use of fertilizers, on average, 33.1 kg ha^−1^ of nitrogen, 48.8 kg ha^−1^ of phosphorus, and 85.9 kg ha^−1^ of potassium, was 3.2 Mg ha^−1^ (Table 1). The highest yield of soybean was obtained on luvisols. Taking into account the previous crop, soybeans sown after root crops had the highest yield. The analysis of yield in the studied seasons revealed that the highest soybean yield of 3.9 Mg ha^−1^ was obtained in 2018, which was characterized by warmer and drier conditions during the soybean vegetation period than normal in Poland (IUNG 2022). Additionally, the autumn and winter of the previous (2017) year were extremely high in precipitation leaving the soil well-hydrated. In the CART analysis, independent variables can be defined as qualitative or quantitative. In Material and Methods, the breakdown of the variables used in the soybean yield analysis into these categories is presented. The variable level shows which categories have been distinguished within one variable, e.g., a variable previous crop has four levels: cereal, legumes, rapeseed, and root crop. Next to the variable level is given its number of observations. The sum of observations in all levels for each variable is 173. Then, the mean and standard deviation is provided for observations in each variable level. 

To determine the contribution of each variable from management and the environment, and to compare the variables’ influence on the yield, plant height, and 1000-grain weight, a regression tree was fitted against all independent variables. In Figure 1, Figure 2 and Figure 3, the optimum regression trees with their splits and final subsets are shown. Importance comparison of independent variables in creating the trees is presented in Figure 4 for all dependent variables, yield, plant height, and 1000-grain weight. 

Variable CWB_10 (climatic water balance from 01 July to 31 August, i.e., from the beginning of the flowering to the full seed stage) was the most important predictor in explaining soybean yield variability. Higher values of CWB_10 (greater than −96 mm, i.e., no drought stress) resulted in a yield of 3.6 Mg ha^−1^, which is around 0.8 Mg ha^−1^ higher than the yield obtained in drier soil conditions (CWB_10 lower than −96 mm). The subset for drier soil conditions and lower values of CWB_10 was further divided by the variable P_2_O_5_ fertilization rate (Figure 1 subset ID 2). For higher P_2_O_5_ fertilization rates (greater than 93.5 kg ha^−1^), grain yield was much higher in comparison to lower phosphorus rates. The subset for phosphorus doses lower than 93.5 kg ha^−1^ and lower yield (mean 2.7 Mg ha^−1^) was then divided according to soil organic carbon content in the soil. Lower grain yield was observed for fields where soil organic carbon was lower than 3.1%. Then, the subset was divided according to clay content. The yield of soybean sown on soils with clay content lower than 11.3% was lower by more than 0.5 Mg ha^−1^ in comparison to the yield obtained on soils with higher clay content (Figure 1 subset ID 6). The subset with higher yield due to higher doses of phosphorus was not divided further (Figure 1 subset ID 5). For soybean grown in soils with a higher amount of water available for plants, the yield was further divided by variable clay content. The yield of soybean sown in soils with clay content higher than 13.9% was higher by more than 0.5 Mg ha^−1^ in comparison to the yield obtained in soils with lower clay content (Figure 1 subset ID 3). The coefficient of the Pearson correlation between observed and predicted yield (based on the CART model) was 0.62. The most important variables in explaining soybean yield variation were CWB_10 and clay (%). Later in order of importance were CWB_1, soil organic carbon (%), pH, soil group (according to FAO WRB), and CWB_7 (Figure 4).

The variable reducing plant height variability the most was CWB_10 (climatic water balance from 01 July to 31 August), as well as in the case of the yield (Figure 2). In drier soil conditions (CWB_10 below −133 mm) plants were shorter by around 20 cm when compared to the height of plants in soil with more water availability. Then, the subset for lower values of CWB_10 was further divided by the variable P_2_O_5_ fertilization rate (Figure 2 subset ID 2). When the rate of phosphorus was higher than 85 kg ha^−1^, the plants were around 20 cm higher than those with smaller doses of P_2_O_5_. Meanwhile, the subset for higher values of CWB_10 was divided by the next variable related to soil water content, i.e., CWB_7 (climatic water balance from 01 June to 31 July). Lower values of CWB_7 (lower than −156 mm) contributed to the lower height of plants. Higher plants with better water availability were then divided into two subsets by the CWB_13 variable (climatic water balance from 01 August to 30 September). Plants were higher if water availability was higher.

When examining the 1000-grain weight, the variable contributing to variability reduction the most was CWB_1 (climatic water balance for the time period from 01 April to 31 May) (Figure 3). When the soil was too wet, it resulted in a smaller grain mass and the difference was around 20 g. Next, other variables related to the environment divided the subsets. In the case of higher grain mass it was CWB_7 (climatic water balance from 01 June to 31 July). When the soil conditions in this period were drier, the mass of 1000-grain was lower, which was then divided by variable clay content. Soils with lower clay content contributed to higher 1000-grain weight. The last division was made by inoculation. When inoculum was used, 1000-grain weight was higher (Figure 3, ID = 6).

The correlation coefficients between the quantitative independent variables and the dependent variables were very different (Table 2). CWB_10, the variable important in reducing variability in all dependent variables, was the most positively correlated with them. Soil organic carbon was negatively correlated with dependent variables, but only in the case of yield was the correlation significant. 

Clay was positively correlated with yield and soybean height but only with yield was it significant. Nitrogen in soil was negatively correlated with all dependent variables but only the correlation with yield and 1000-grain weight was it significant. Almost all climatic indices (CWB_1 to CWB_13) were positively correlated with dependent variables. In the case of CWB_1, the correlation was negative for yield and 1000-grain weight, but only for 1000-grain weight was it positive. There is also a negative correlation between fertilizer amount and the dependent variables. Only in the case of nitrogen was the correlation positive for yield and height. Sowing also had a positive effect on all dependent variables, i.e., the later the sowing was performed, the higher values of variables were observed. A positive correlation was also observed between pesticides and dependent variables.

The regression analysis showed that in the case of all dependent variables, the most statistically significant was CWB_10. The regression coefficient for CWB was positive (i.e., the yield, height, and 1000-grain weight increased with the increase of CWB_10). The coefficients of determination for regression were R^2^ = 0.40 for yield, R^2^ = 0.34 for height and R^2^ = 0.20 for 1000-grain weight (Table 3).

## 3. Discussion

The results from CART showed that soybean yield variability was mostly dependent on the soil’s water content in the time period from 01 July to 31 August (represented by the variable CWB_10), i.e., from the beginning of flowering to full seed stage for most of the plants. It confirms that soybean yield in rain-fed conditions is highly dependent on seasonal water supply during reproductive stages. Availability of water for plants in earlier growth stages (variables: CWB_1, CWB_4, and CWB_7) had a weaker effect on grain yield but was still significant. Such results were obtained in the study of Zanon et al. [20] conducted in subtropical environments. The results obtained in our study are consistent with those of Eck et al. [25] in which water stress during various reproductive stages (from R1 to R6) reduced grain yield by 9 to 65%. The highest reduction of the yield was observed when water stress occurred between stages R5 (beginning seed) and R6 (full seed). The effect of water stress during seed filling can be stronger if water stress occurs together with heat stress. These two stresses can cause a very high decrease in the grain yield of soybean; however, these stresses have a non-significant effect on seed weight. The result is opposite to our study because in this study both variables (grain yield and 1000-grain weight) were positively correlated.

For many crops grown in Poland, the environment is responsible for most of the total experimental variation in grain yield [26,27,28,29]. In the case of rain-fed soybean production, water scarcity, as well as excess water, may be limiting factors. Unfortunately, weather conditions in Central and Eastern Europe in different years can be completely different, from severe drought to rain during summer, when the reproductive stages of soybean occur. 

The analysis also resulted in a conclusion that the soybean yield depends on agronomic management practices (especially phosphorus fertilization) but also soil fertility (especially clay content) for soybean growth. This outcome is in line with the findings of Zheng et al. [30] in China, who recommend that government programs should pay close attention to the management practices of farmers to prevent ineffective management. Usually, nitrogen fertilization is most important for most crops; however, for soybean, a very important nutrient is phosphorus. In this study, phosphorus fertilization discriminates grain yield of soybean which is confirmed by many studies [31,32].

The date of sowing was an important factor in explaining soybean yield variability; however, the effect of the sowing date was much weaker in comparison to environmental factors. Before sowing, the temperature and moisture level of the soil should be checked. When the soil is too cold, plant emergence is delayed and the seeds are exposed to pathogens. Frosts can also increase plant losses. Other variables were also observed as important in explaining the variability of the yield of soybean grown in Poland, such as potassium fertilization [33,34,35].

Soybeans are sensitive to sandy soils, especially with low soil reaction. It was confirmed in this study because the correlation between soil pH and grain yield was positive and statistically significant. Sowing in sites with too low of a pH interferes with the nodulation process. Under such conditions nitrogen-fixing bacteria are unable to develop. The optimum pH of the soil for soybean cultivation is in the range of 6–6.5. In our experiments, the pH had an average value of 6.3.

As shown by numerous studies, for soybean there is a compromise between the biological N_2_ fixation and the nitrogen supply in the soil, i.e., the fixation decreases with increasing nitrogen content in the soil and vice versa [36]. As a result, the application of nitrogen fertilizer reduces N_2_ fixation, which results in an increase in nitrogen uptake [37]. The proportion of nitrogen utilization from both of these sources varies with environmental and soil conditions, depending on, for e.g., temperature, soil moisture, or soil pH. Biological N_2_ fixation in soybean represents, on average, 60% of total nitrogen uptake. In the experiment consisting of a control with zero nitrogen and a fertilized treatment with 600 kg ha^−1^ of nitrogen applied to four soybean genotypes, the control and fertilized treatments had different nitrogen accumulation from the biological N_2_ fixation, but there were no differences observed in seed yield [36]. There is evidence of weak soybean yield response to soil nitrogen fertilization broadcasted on the surface. However, when the biological N_2_ fixation was unable to meet nitrogen needs, nitrogen fertilization may overcome nitrogen limitations [37]. La Menza et al. [2] compared treatment with zero nitrogen, forcing the crop to rely on biological N_2_ fixation, with full nitrogen treatment designed to optimally match the expected seasonal nitrogen requirements of plants. The researchers found that the average seed yield was higher in the full nitrogen than in the zero nitrogen treatment [38]. Kaschuk et al. [39] stated that *Bradyrhizobium* inoculation is sufficient to meet all soybean nitrogen needs and that nitrogen fertilizer negatively affects the number of nodules and dry weight indicating a reduction in symbiotic efficiency. Thus, there are still discussions about whether increasing nitrogen fertilization will increase the seed yield and whether it is profitable [37,40,41]. Such studies regarding soybean nitrogen requirements have never been conducted in Poland, as soybean cultivation is still not widespread. However, it is not known why the higher nitrogen doses negatively affected the soybean yield. On the basis of the available data, it was not determined whether this was due to a reduction in the number of nodules or other factors. However, some nitrogen fertilization is necessary to overcome nitrogen deficiency at the beginning of vegetation of soybean as a starter which has the potential to increase soybean yield [42]. The regression model results suggest that a safe nitrogen application rate would be probably between 30 and 45 kg ha^−1^. The analysis presented in this study could form the basis for more targeted research in the future. 

In addition to the advantages of growing soybeans mentioned in the introduction, another great advantage is its beneficial effect on the soil. Soybeans improve the physicochemical properties and soil fertility [43]. Soybean as a legume previous crop very positively affects the next crops, especially wheat, which is very important in sustainable crop rotation [44]. It is especially economically justified because of the very high prices of nitrogen fertilizers. For better recognition of optimal growing conditions for soybeans in Poland, it might be necessary to conduct more experiments with more diverse doses of fertilizers and pesticides and detailed recognition of environmental growth conditions.

One of the limitations of the presented study is that the data used in this research comes from an experiment intended for a varietal assessment. Therefore, there was no part of the relevant soil data that was replaced with modeled data. However, this research provides the basis for future analysis as it covers many locations with different soil and weather conditions and varied management. The study confirmed that environmental variables are more important in the discrimination of yield and yield-related traits variability in comparison to crop management factors, which had a much smaller effect. Especially meteorological variables (CWB) and natural soil properties, such as soil texture and content of soil organic carbon, had the highest effect on yield variability. The effect of agronomical practices on grain yield and the height of soybean plants was relatively small, but slightly higher for the mass of grains.

## 4. Materials and Methods

### 4.1. Experimental Design

To assess the causes of variability in soybean yield, plant height, and 1000-grain weight, the variables related to crop management and the environment were used. The soybean data came from the Polish Post-Registration Variety Testing System [45]. The most frequently occurring cultivars together with their mean grain yield used in the study are presented in Appendix A. The experiments were conducted during 10 growing seasons (harvest years 2012–2021) in 37 locations (Figure 5) and 85 modern high-yield varieties of soybean were used. For each level of quantitative environmental and management variables, means and standard deviations of the yield, plant height and 1000-grain weight were presented (Table 4). 

Soybean was sown at the end of April or the beginning of May and harvested at the end of September or the beginning of October. In each experiment, the sowing density was 70 seeds m^−1^ which provides an optimal plant density. Soybeans were sown in a row spacing of 20 cm. The soybean yield was expressed as grain yield in Mg per hectare, height was measured in cm, and 1000-grain weight was provided in g. These were the dependent variables. The determinants of soybean yield and yield-related traits used for the construction of classification and regression trees (CART) were variables related to the soil from the soilgrids.org website, available N (g kg^−1^), soil organic carbon (%), sand (%), clay (%), and soil group FAO WRB [46]. The pH was measured in each experiment. Crop management factors included the rate of nitrogen, phosphorus, potassium expressed in kg ha^−1^ of pure nutrient, the number of herbicides, expressed in l ha^−1^, of active substance, the number of insecticides, expressed in kg ha^−1^, of active substance, the use fungicides, expressed as yes or no, regarding their application, inoculum, expressed as yes or no, regarding its application, sowing date, and previous crop. The Institute of Soil Science and Plant Cultivation-State Research Institute (IUNG-PIB) in Puławy determines the values of climatic water balance (CWB), which were used in this study as an environmental variable that characterizes soil moisture [47]. CWB is expressed in millimeters of water and, in the conditions of the Polish climate, it can vary from ca. -240 (extra severe drought) to 50 (excess water). Selected periods for which CWB was calculated for the most important growth phases of soybean are presented in Table 5. 

Soybean growing conditions were determined for each location and the average yield, plant height, and 1000-grain weight were calculated for all cultivated varieties. Normally, in those kinds of trials, a genetic variable was not that important in explaining, for example, yield variability. The reason is that the multi-environmental trials are using modern varieties to obtain a high, stable yield [26,27,28,29]. All three dependent variables were measured at 14–15% grain moisture. The field experiments (observations) were carried out as a one-factorial design (variety was the factor) with four replicates. The area of a single plot was 16.5 m^2^. In each location, conventional tillage was used. The agricultural management included nitrogen, phosphorus and potassium fertilization, the use of herbicides, insecticides and fungicides, and the use of inoculum (symbiotic bacteria which fix atmospheric nitrogen). The average protein content was 36% and the average oil content was 23%. Unfortunately, these measurements were rarely performed and there is no sufficient number of observations that can be used for a comprehensive analysis. If the soybean dataset used in this study was compared to a complete L × Y (location and year) classification, that would give 47% of possible combinations. The number of observations for each level of variables is shown in Table 1. 

### 4.2. Statistical Analysis

Multiple linear regression analyses were applied to examine the influence of quantitative independent variables on soybean yields, where the following linear model was used:(1)$$y_{i}=\left(\beta_{0}+\overset{n}{\underset{i=1}{\sum }}\beta_{i}\cdot x_{i}\right)+e_{i}$$
where: $y_{i}$ is the soybean yield (in kg) and $x_{i}$ is the quantitative independent variable.

Due to the unbalanced nature of the dataset, the analysis of Polish soybean yield relationships and interactions with environmental, agronomic management, and genetic variables was performed using the CART model [48,49]. The importance of the variables in determining the yield, plant height, and-1000 grain weight was compared using CART [48]. A 10-fold cross-validation method was used to “prune” overgrowth trees. Analysis was performed with STATISTICA software ver. 13 [50]. Classification and Regression Trees or CART is a term introduced to refer to Decision Tree algorithms that can be used for regression predictive modeling problems. The algorithm of decision tree models works by repeatedly partitioning the data into multiple subsets, so that the outcomes in each final subset are as homogeneous as possible. Splitting criteria are based on variance-minimizing algorithms. The response of the dependent variable is explained by a set of independent continuous or categorical variables. The produced result for regression trees consists of a set of rules used for predicting the outcome continuous variable [51]. Here, the dependent variable was soybean yield, and the independent variables were the treatments and parameters investigated within the field trials. Based on the optimal tree, the importance of each independent variable in the final CART tree can be calculated [26,51].

## 5. Conclusions

Soybean is a multi-purpose crop in which seeds are rich in nutrients. Soybean is also a very good previous crop as it leaves the soil naturally drained and loosened with significant amounts of nitrogen. However, soybeans have particularly high thermal demands. Thus, soybean cultivation in Poland is associated with the risk of plant damage due to too low temperatures. Considering future climate change, Poland might play a bigger role in soybean production. Findings from this study can serve as an introduction to extended research to guide nitrogen and other agronomic management in the successful growth of soybean in the hemiboreal climates in Poland and other Central and Eastern European countries. The most important factors which have the strongest effect on grain yield and yield-related traits of soybean were environmental conditions (water availability, clay content, and soil organic carbon). The most important variable which discriminated yield and yield-related traits was climatic water balance during reproductive stages. Crop management had a weaker effect on the yield. Phosphorus was the most important nutrient, of which fertilization increased the grain yield of soybean. Moreover, the positive effect on the grain yield of soybean had soil pH. Mineral fertilization and crop protection have relatively low effects on grain yield and yield-related traits of soybeans. The grain yield of soybean was positively correlated with the height of plants and grain weight. Production of soybean can be of great importance to the Central and Eastern European agricultural economy and the evaluation of factors that determine grain yield variability is very important for future crop production and plant breeding.

## Figures and Tables

**Figure 1 plants-11-02922-f001:**
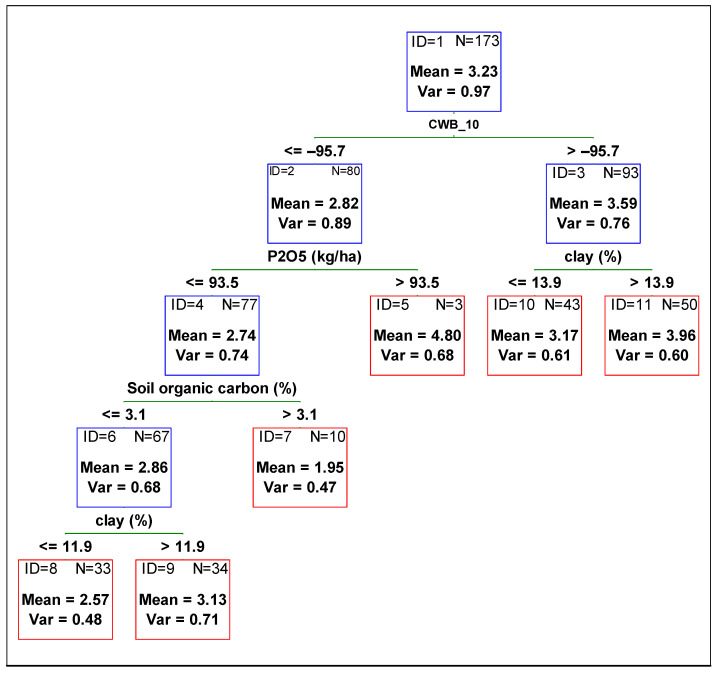
Regression tree predicting soybean yield from agronomic management and environmental variables. Red boxes indicate final subsets and the blue ones indicate subsets that were divided. CWB_10 time period from 01 July to 31 August.

**Figure 2 plants-11-02922-f002:**
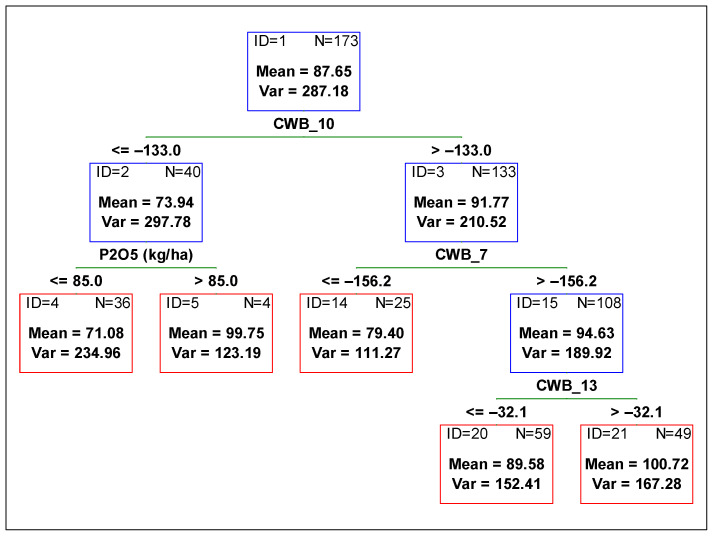
Regression tree predicting soybean plants height [cm] from agronomic management and environmental variables. Red boxes indicate final subsets and the blue ones indicate subsets that were divided. CWB_7 time period from 01 June to 31 July, CWB_10 from 01 July to 31 August, and CWB_13 from 01 August to 30 September.

**Figure 3 plants-11-02922-f003:**
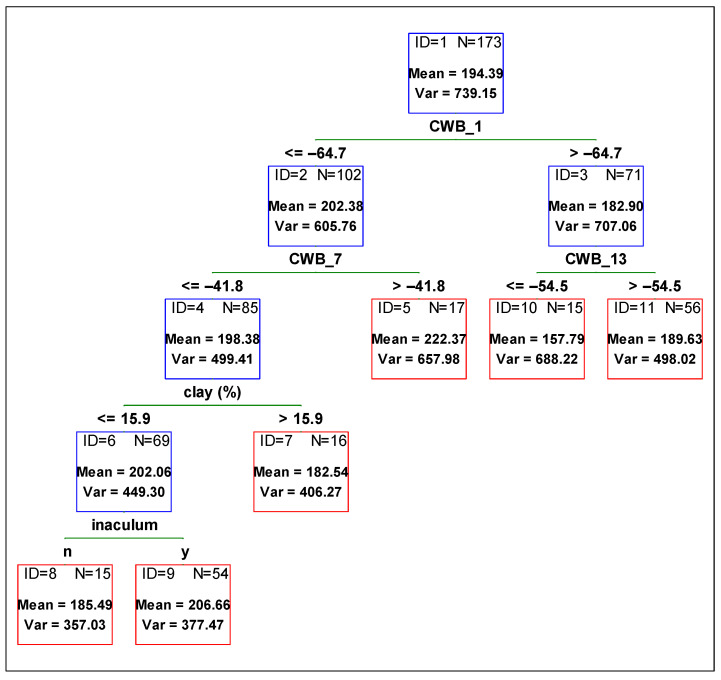
Regression tree predicting soybean 1000-grain weight [g] from agronomic management and environmental variables. Red boxes indicate final subsets and the blue ones indicate subsets that were divided. CWB_1 time period from 01 April to 31 May, CWB_7 from 01 June to 31 July, and CWB_13 from 01 August to 30 September.

**Figure 4 plants-11-02922-f004:**
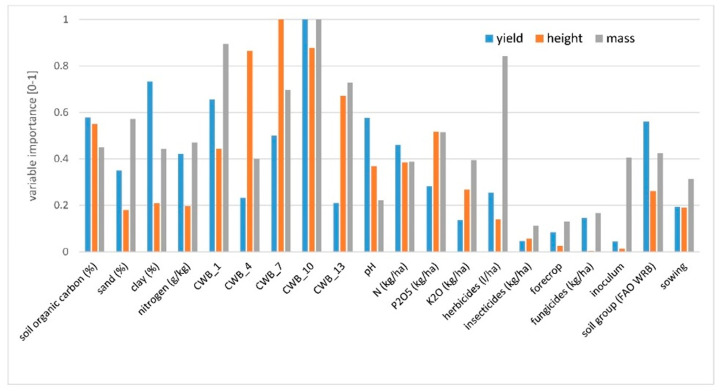
Importance of independent variables on the scale of 0–1 (importance 0—variable is never featured in the tree, importance 1—the highest importance in the explanation of dependent variable variability) for yield, plant height, and 1000-grain weight in soybean cultivation. CWB_1 time period from 01 April to 31 May, CWB_4 from 01 May to 30 June, CWB_7 from 01 June to 31 July, CWB_10 from 01 July to 31 August, and CWB_13 from 01 August to 30 September.

**Figure 5 plants-11-02922-f005:**
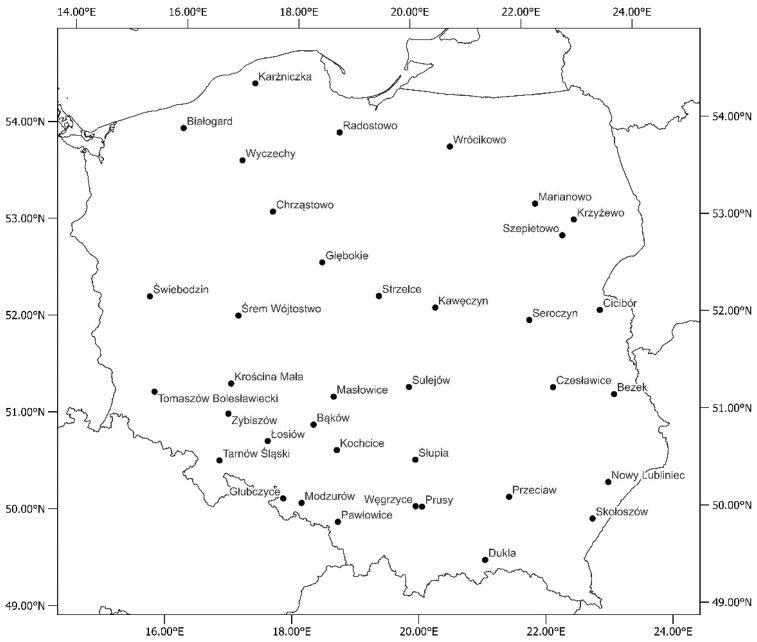
Locations of the soybean experiments which took place between 2012 and 2021 in Poland.

**Table 1 plants-11-02922-t001:** Mean ± SD of dependent variables yield, height of plants, and 1000-grain weight and for independent quantitative variables used in the study.

		Mean ± SD	Min–Max
independent quantitative	Soil organic carbon (%)	2.0 ± 0.6	1.2–3.6
	sand (%)	50.2 ± 20.4	11.6–79.0
	clay (%)	15.5 ± 7.5	5.3–27.3
	Soil nitrogen (g kg^−1^)	1.8 ± 0.5	1.1–3.0
	CWB_1	−75.6 ± 60.5	−188.3–130.6
	CWB_4	−93.0 ± 69.6	−227.7–121.4
	CWB_7	−106.3 ± 68.4	−254.7–62.9
	CWB_10	−74.5 ± 77.5	−231.3–175.0
	CWB_13	−40.1 ± 65.9	−173.4–123.0
	Soil pH	6.3 ± 0.5	4.0–7.6
	rate of N (kg ha^−1^)	33.1 ± 18.1	0.0–105.0
	rate of P_2_O_5_ (kg ha^−1^)	48.8 ± 21.9	0.0–105.0
	rate of K_2_O (kg ha^−1^)	85.9 ± 29.5	0.0–163.0
	herbicides (l ha^−1^)	3.3 ± 2.1	0.0–9.0
	insecticides (kg ha^−1^)	0.1 ± 0.2	0.0–1.9
dependent	yield (Mg ha^−1^)	3.2 ± 1.0	1.0–5.4
	plant height (cm)	87.6 ± 17.0	38.0–121.0
	1000-grain weight (g)	194.4 ± 27.3	111.8–261.9

**Table 2 plants-11-02922-t002:** Pearson correlation coefficients for between quantitative variables used in the study with grain yield and yield-related traits (significant correlations at 0.05 significance level are marked in red).

	Yield (Mg ha^−1^)	Height (cm)	1000-Grain Weight (g)
soil organic carbon (%)	−0.220	−0.055	−0.136
sand (%)	−0.202	−0.096	0.109
clay (%)	0.301	0.102	−0.091
Soil nitrogen (g kg^−1^)	−0.192	−0.035	−0.158
CWB_1	−0.004	0.144	−0.246
CWB_4	0.050	0.295	−0.003
CWB_7	0.258	0.427	0.112
CWB_10	0.293	0.442	0.046
pH	0.180	0.066	−0.049
CWB_13	0.068	0.311	−0.088
N (kg ha^−1^)	0.100	0.161	−0.053
P_2_O_5_ (kg ha^−1^)	0.015	−0.038	−0.106
K_2_O (kg ha^−1^)	−0.102	−0.026	−0.109
sowing	0.123	0.009	0.050
herbicides (l ha^−1^)	0.119	0.008	0.191
insecticides (kg ha^−1^)	0.024	0.029	0.004
yield (Mg ha^−1^)		0.427	0.396
height (cm)			−0.009

**Table 3 plants-11-02922-t003:** Multivariate linear regression coefficients (b) and *p*-values. The dependent variables were yield, height, and 1000-grain weight (in red are marked coefficients and *p*-values which indicate significant relationships at 0.05 probability level).

	Yield	R^2^ = 0.40	Height	R^2^ = 0.34	1000-Grain Weight	R^2^ = 0.20
	b	*p*-Value	b	*p*-Value	b	*p*-Value
y-intercept	−2.050	0.131	68.162	0.005	214.266	0.000
soil organic carbon (%)	−0.078	0.603	0.837	0.756	0.031	0.995
sand (%)	0.033	0.000	0.178	0.271	0.096	0.738
clay (%)	0.131	0.000	0.262	0.549	0.285	0.713
Nitrogen (g kg^−1^)	−0.210	0.283	−1.745	0.619	−8.555	0.170
CWB_1	−0.003	0.118	−0.021	0.532	−0.194	0.001
CWB_4	0.000	0.987	0.064	0.079	0.144	0.026
CWB_7	0.001	0.796	0.038	0.318	−0.079	0.244
CWB_10	0.006	0.002	0.084	0.009	0.133	0.020
CWB_13	−0.002	0.201	0.017	0.595	−0.079	0.166
pH	0.286	0.024	3.152	0.163	−1.520	0.703
sowing	0.007	0.268	0.121	0.301	−0.005	0.982
N (kg ha^−1^)	0.006	0.095	0.117	0.079	−0.134	0.258
P_2_O_5_ (kg ha^−1^)	0.001	0.742	−0.020	0.743	−0.139	0.195
K_2_O (kg ha^−1^)	−0.002	0.510	−0.023	0.631	−0.040	0.631
herbicides (l ha^−1^)	0.080	0.012	0.161	0.776	2.398	0.018
insecticides (kg ha^−1^)	−0.088	0.824	−1.781	0.801	−4.116	0.742

**Table 4 plants-11-02922-t004:** Mean ± SD of yield, the height of plants, and 1000-grain weight for different combinations of independent qualitative variables used in the study.

			Yield (Mg ha^−1^)		Height (cm)		1000-Grain Weight (g)	
		n	Mean ± SD	Min–Max	Mean ± SD	Min–Max	Mean ± SD	Min–Max
**Soil group (FAO WRB)**	Albeluvisols	17	3.1 ± 0.7	1.4–4.2	94.1 ± 12.6	74.4–118.0	186.1 ± 20.2	149.4–212.0
	Cambisols	40	2.8 ± 1.1	1.0–5.0	83.8 ± 19.5	38.0–121.0	194.4 ± 26.5	148.0–257.0
	Fluvisols	4	2.5 ± 0.5	2.1–3.0	73.8 ± 8.4	66.0–81.0	182.8 ± 16.5	168.0–197.0
	Gleysols	10	3.3 ± 0.4	2.9–4.0	98.6 ± 12.4	83.0–112.0	198.2 ± 22.0	166.4–230.1
	Luvisols	102	3.5 ± 1.0	1.0–5.4	87.5 ± 16.5	39.0–121.0	195.9 ± 29.3	111.8–261.9
**previous crop**	cereal	146	3.2 ± 0.9	1.0–5.4	87.7 ± 17.7	38.0–121.0	193.5 ± 27.2	111.8–257.0
	legumes	3	3.2 ± 0.7	2.6–4.0	94.3 ± 12.9	80.0–105.0	193.4 ± 36.5	151.3–217.0
	rapeseed	9	3.2 ± 1.1	1.6–4.6	87.7 ± 12.2	71.0–106.0	204.0 ± 25.4	163.3–242.7
	root crop	15	3.6 ± 1.4	1.3–5.4	85.9 ± 12.9	66.0–105.0	197.6 ± 29.0	150.8–261.9
	inoculum	138	3.3 ± 1.0	1.0–5.4	87.5 ± 17.8	38.0–121.0	196.0 ± 27.7	111.8–257.0
	no inoculum	35	3.1 ± 0.9	1.2–5.4	88.2 ± 13.6	54.7–110.0	187.9 ± 24.8	141.0–261.9
	no fungicides	170	3.2 ± 1.0	1.0–5.4	87.6 ± 17.1	38.0–121.0	193.9 ± 26.7	111.8–257.0
	fungicides	3	4.3 ± 1.8	2.2–5.4	91.3 ± 11.7	81.0–104.0	219.5 ± 51.2	162.7–261.9
**year**	2012	4	2.9 ± 0.6	2.1–3.6	95.3 ± 4.5	93.0–102.0	155.4 ± 25.7	118.6–175.3
	2013	4	2.6 ± 1.3	1.0–3.8	84.0 ± 4.2	81.0–90.0	150.3 ± 16.9	135.0–173.9
	2014	5	3.4 ± 0.8	2.5–4.5	108.0 ± 15.2	82.0–121.0	199.4 ± 11.5	180.6–211.0
	2015	6	2.2 ± 1.0	1.0–3.3	68.6 ± 21.3	38.0–100.0	151.7 ± 21.4	111.8–175.7
	2016	11	3.2 ± 0.7	2.2–4.7	92.1 ± 12.7	73.0–106.0	178.2 ± 26.8	151.3–224.1
	2017	10	3.3 ± 0.7	2.0–4.5	90.1 ± 14.2	75.0–112.0	188.3 ± 22.4	139.0–215.0
	2018	30	3.9 ± 0.9	2.5–5.4	85.0 ± 13.7	53.0–108.0	210.4 ± 20.2	163.3–261.9
	2019	30	2.9 ± 1.0	1.2–4.9	71.2 ± 14.4	39.0–95.0	201.7 ± 21.5	164.6–243.0
	2020	35	3.1 ± 0.9	1.4–4.9	89.6 ± 14.9	62.0–118.0	208.1 ± 24.9	161.0–257.0
	2021	38	3.3 ± 1.0	1.4–5.2	98.9 ± 12.7	77.4–121.0	184.5 ± 21.1	141.0–228.0

**Table 5 plants-11-02922-t005:** Periods and growth phases of soybean for which climatic water balance (CWB) is calculated.

Abbreviation Used for Certain Climatic Water Balance	Period for which CWB Is Calculated	Growth Phase of Soybean for which CWB Is Calculated *
CWB_1	from 01 April to 31 May	from before sowing to the emergence of soybean
CWB_4	from 01 May to 30 June	from sowing to the end of vegetative stage
CWB_7	from 01 June to 31 July	from the emergence to the full flowering
CWB_10	from 01 July to 31 August	from the beginning of the flowering to the full seed stage
CWB_13	from 01 August to 30 September	from the beginning of pod stage to the full maturity

* Typical growth phases are presented for most of the plants, but they can vary depending on location and because of the variability between cultivars and plants within the experiments.

## Data Availability

Not applicable.

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
