# Peer review of "Analysis of the Impact of Environmental and Agronomic Variables on Agronomic Parameters in Soybean Cultivation Based on Long-Term Data"

_plants, 2022, doi:10.3390/plants11212922_

Round 1
Reviewer 1 Report
General comment:
The study is devoted to the historical analysis of some characteristics of soybean under the influence of mineral fertilization, herbicides, sowing date, and the preceding crop. Long-term data from the Polish Post-Registration Variety Testing System in multi-environmental trials were used. Cultivar evaluations depend on data generated from field trials, based on different levels of inputs. However, soybean cultivars should fulfill multiple requirements, including other agronomic performance and quality. This data synthesis can help to take advantage of existing data for the purpose of decision-making, but the proposals are necessary. Basically, there is no concrete conclusion, on which cultivar the authors recommend and with which inputs for its cultivation. As is known, cultivars are area-specific with regard to optional adaptation. The number of related arguments and conclusions raises questions. English requires improvement.

Author Response
General comment:
The study is devoted to the historical analysis of some characteristics of soybean under the influence of mineral fertilization, herbicides, sowing date, and the preceding crop. Long-term data from the Polish Post-Registration Variety Testing System in multi-environmental trials were used. Cultivar evaluations depend on data generated from field trials, based on different levels of inputs. However, soybean cultivars should fulfill multiple requirements, including other agronomic performance and quality.
This data synthesis can help to take advantage of existing data for the purpose of decision-making, but the proposals are necessary.
Basically, there is no concrete conclusion, on which cultivar the authors recommend and with which inputs for its cultivation.
As is known, cultivars are area-specific with regard to optional adaptation.
The number of related arguments and conclusions raises questions.
English requires improvement.
Authors: Thank you very much for all the comments which allow as to improve the manuscript. The manuscript was corrected according to all of the comments and we hope that current version fulfill all the requirements.
The Abstract is structured with a background, the main body of the abstract, and a short conclusion. However, the Abstract is not very enlightening – be more insightful about the findings. I would suggest more details in the abstract (influence of sowing date and preceding crop).
Authors: The Abstract was extended and more details were added, including effect of crop management (e.g. sowing date and preceding crop)
General comment to the Introduction section:
The content of the literature review chapter is related to the research topic. Up-to-date literature references are presented in the manuscript by the author(s), but one should look at the critical analysis of the influence of the investigated factors on the variables of soybean. A good rule of thumb is to use sources published in the past 10 years for research.
Authors: The Introduction was extended, especially the influence of the investigated factors on the studied variables was better described.
In the chapter "Results",
The results are displayed correctly. The tables and figures are clear.
Authors: Thank you very much for your positive opinion
In the chapter "Discussion",
The discussion is informative. The authors attempt to discuss their important results and the rest is a quotation from the literature. Please try to discuss your results in depth and link them with the appropriate literature throughout the whole section. In my opinion, the most interesting conclusion is that investigated traits at different genotypes reacted differently to changes in factors.
Authors: Discussion was improved and extended.
Lines 208-2010. How do you explain this phenomenon considering that the seed yield was positively and significantly correlated with 1000-seed weight?
Authors: The results in which water stress and heat stress have negative effect on grain yield but not on seed weight are from not our study. We have added in the Discussion following sentence: The result is opposite to our study because in this study both variables (grain yield and seed weight) were positively correlated.
Lines 234-262. Many studies demonstrate that applying N as a starter has the potential to increase soybean yield.
Authors: Such information was added to the Discussion
It is necessary to encourage further interest in the cultivation and expansion of soybeans and to show the relevance of further investments based on the decisions made.
Authors: Such information was added to the Discussion
In the chapter "Materials and Methods", the methodology is adequate but lacks information in some aspects. The information about genotypes should be explained in the text. Seed yield evaluation has been done using the complete plot so that the border effect was not avoided.
Authors: The explanation about genotypes is in Material and Methods. Moreover, we added Supplementary Table 1S with most frequently used cultivars in the experiments and their grain yield. Cultivars used in the multi-environmental trials often are area-specific with regard to optional adaptation. These are highly productive unified cultivars.
In the chapter "Conclusions”
Highlight the influence of other factors relevant to the research.
Authors: It has been added to the Conclusions
Line 14. livestock
Line 16. This work used a…
Line 17. from the years 2012-2021
Line 18. What mineral fertilizers (N, P, K) were used?
Line 19. Change “type of forecrop” to “preceding crop” or “previous crop”.
The environment..
Line 21. models
Line 25. Clear “the” (on the soybean); climate
Line 26. yield
Line 34. average of 4.5 Mg ha−1
Table 1. Change “forecrop” to “preceding crop” or “previous crop”.
Authors: All the corrections have been made
Reviewer 2 Report
As stated by the authors, soybeans have not received much attention as a profitable crop in Poland due to sub-optimal growing conditions and little economic importance compared to other crop choices. However, interest among Polish farmers is there and basic agronomic research needs to be done to generate data farmers can use. Thus, the authors analyzed data from a multi-year variety trial to establish a baseline to guide future studies.
I agree with the authors in accepting this study as a useful introduction to promote additional research of soybeans in Poland. The methods seemed appropriate in making use of available, though somewhat incomplete, data from the variety trials. The CART results allow a relatively straight-forward interpretation of the variables impacting soybean yield, which unfortunately, are largely up to the whims of weather.
I do have several criticisms that need to be addressed. The main one has to do with organization of the main sections. Results come before the Methods, and the tables and figures are out of order which makes for difficult reading. I also suggest reorganizing subheadings in Table 1 to yield, height, 1000 grain weight to be consistent with other tables. Moreover, the first paragraph of the Results could be edited to describe Tables 1 & 3 first followed by the findings (currently the opposite).
Could the regression trees be explained better or figures improved to make the interpretations clearer to those not used to looking at these? Perhaps, but I don't have any specific recommendations. Maybe emphasizing the take home message or main finding somehow in each tree?
Author Response
As stated by the authors, soybeans have not received much attention as a profitable crop in Poland due to sub-optimal growing conditions and little economic importance compared to other crop choices. However, interest among Polish farmers is there and basic agronomic research needs to be done to generate data farmers can use. Thus, the authors analyzed data from a multi-year variety trial to establish a baseline to guide future studies.
I agree with the authors in accepting this study as a useful introduction to promote additional research of soybeans in Poland. The methods seemed appropriate in making use of available, though somewhat incomplete, data from the variety trials. The CART results allow a relatively straight-forward interpretation of the variables impacting soybean yield, which unfortunately, are largely up to the whims of weather.
Authors: Thank you very much for the review and positive opinion.
I do have several criticisms that need to be addressed. The main one has to do with organization of the main sections. Results come before the Methods, and the tables and figures are out of order which makes for difficult reading. I also suggest reorganizing subheadings in Table 1 to yield, height, 1000 grain weight to be consistent with other tables. Moreover, the first paragraph of the Results could be edited to describe Tables 1 & 3 first followed by the findings (currently the opposite).
Authors: All these required changes have been included in the revised manuscript but the Methods were left after the Results because it is required by the journal in guidelines for authors.
Could the regression trees be explained better or figures improved to make the interpretations clearer to those not used to looking at these? Perhaps, but I don't have any specific recommendations. Maybe emphasizing the take home message or main finding somehow in each tree?
Authors: The description of the results was extended to make the interpretations clearer.